# Optimal Prandial Timing of Insulin Bolus in Youths with Type 1 Diabetes: A Systematic Review

**DOI:** 10.3390/jpm12122058

**Published:** 2022-12-13

**Authors:** Enza Mozzillo, Roberto Franceschi, Francesca Di Candia, Alessia Ricci, Letizia Leonardi, Martina Girardi, Francesco Maria Rosanio, Maria Loredana Marcovecchio

**Affiliations:** 1Department of Translational Medical Science, Section of Pediatrics, Regional Center of Pediatric Diabetes, Federico II University of Naples, 80131 Naples, Italy; 2Pediatric Diabetology Unit, Pediatric Department, Santa Chiara General Hospital of Trento, 38122 Trento, Italy; 3Department of Pediatrics, University of Cambridge and Cambridge University Hospitals NHS Foundation Trust, Cambridge CB2 0QQ, UK

**Keywords:** insulin bolus, timing, post-prandial blood glucose

## Abstract

The aim of this systematic review was to report the evidence on optimal prandial timing of insulin bolus in youths with type 1 diabetes. A systematic search was performed including studies published in the last 20 years (2002–2022). A PICOS framework was used in the selection process and evidence was assessed using the GRADE system. Up to one third of children and adolescents with type 1 diabetes injected rapid-acting insulin analogues after a meal. Moderate–high level quality studies showed that a pre-meal bolus compared with a bolus given at the start or after the meal was associated with a lower peak blood glucose after one to two hours, particularly after breakfast, as well as with reduced HbA1c, without any difference in the frequency of hypoglycemia. There were no differences related to the timing of bolus in total daily insulin and BMI, although these results were based on a single study. Data on individuals’ treatment satisfaction were limited but did not show any effect of timing of bolus on quality of life. In addition, post-prandial administration of fast-acting analogues was superior to rapid-acting analogues on post-prandial glycemia. There was no evidence for any difference in outcomes related to the timing of insulin bolus across age groups in the two studies. In conclusion, prandial insulin injected before a meal, particularly at breakfast, provides better post-prandial glycemia and HbA1c without increasing the risk of hypoglycemia, and without affecting total daily insulin dose and BMI. For young children who often have variable eating behaviors, fast-acting analogues administered at mealtime or post-meal could provide an additional advantage.

## 1. Introduction

Post-prandial hyperglycemia is a key factor influencing glycemic outcomes in children and adolescents with type 1 diabetes (T1D) [1]. There are currently three rapid-acting insulin analogues on the market, and manufacturers recommend injecting insulin five to 10 minutes prior to a meal (Aspart) or up to 15 to 20 minutes after a meal (Lispro and Glulisine, respectively) [2,3,4]. In adults, pharmacokinetic and pharmacodynamic studies of rapid-acting insulin analogues [5] and continuous glucose monitoring (CGM) systems data [6,7] showed that the injection of insulin 10 to 30 minutes before a meal provides optimal post-prandial glycemia [8]. Other studies in adults showed that the post-prandial spike is more effectively controlled by proper timing of insulin administration rather than increasing the pre-meal insulin dose or administering a super-sized correction bolus, which could result in hypoglycemia [9]. Inaccurate insulin bolus timing has been shown to result in suboptimal glycemic control in people with T1D [1]. Fast-acting analogues are now available on the market, and they have an onset of action approximately five to seven minutes earlier, and a glucose lowering effect 78–147% higher, than rapid-acting analogues [10], and they should provide more flexibility in the timing of the insulin bolus.

Timing of bolus is critical even when advanced insulin technologies are used. Data from advanced hybrid closed loop (AHCL) systems clearly show that the timing of bolus remains critical to achieve optimal glycemic targets, and a delayed administration may cause automated over-delivery of insulin and subsequent hypoglycemia [11]. In fact, even with the use of algorithms, refinements to mealtime boluses are necessary in order to control prandial glycemic excursions [11]. Smart pens, which record and store data on the amount and timing of recent insulin injections, provide dose reminder alerts, and the option to view active insulin on board, may facilitate and improve diabetes management and support people with T1D in achieving better timing of insulin boluses, particularly if combined with CGM use [12]. However, there is not enough supportive strong evidence for any of the bolus timing strategies, likely because significant interindividual and intraindividual variations exist in post-prandial glucose peak times [13]. A previous review on optimal prandial timing of bolus insulin concluded that in adults with T1D, the administration of rapid-acting insulin analogues 15–20 min before a meal led to a 30% reduction in post-meal glucose levels and rates of hypoglycemia compared with a bolus given immediately before a meal [8]. Improving post-meal glycemia is important as this will result in better HbA1c levels [14].

In the pediatric population, rapid-acting insulin analogues are now widely used, but there is no clear consensus on the optimal timing of bolus and whether this varies according to age. Given the recent introduction of fast-acting analogues, it would also be important to have an overview of the evidence of the optimal timing of bolus for these new insulin formulations. The aim of this systematic review is to provide an up-to-date summary of the available evidence on optimal prandial timing of insulin boluses in the pediatric population with T1D and its effect on glycemic outcomes and on treatment satisfaction.

## 2. Material and Methods

### 2.1. Search Strategy

Data for the present review were collected through searches of Pubmed, EMBASE, the Cochrane Library, Web of Science, Clinicaltrial.gov, and the International Clinical Trials Registry Platform. Articles published between 1 January 2002 and 30 September 2022 were considered. Search terms, or “MESH” (Medical Subject Headings) used different combinations of terms: “insulin timing” or “time of dose” or “timing of bolus” or “timing of prandial” or “insulin-meal interval” AND “Type 1 diabetes” or “T1D” “insulin dependent” AND “post-prandial hyperglycemia” or “post-prandial excursion” or “metabolic control” or “glucose level” or “hypoglycemia” or “total daily insulin” or BMI or “treatment satisfaction”.

### 2.2. Criteria for Study Selection

We conducted a systematic search of the literature according to the PICOS model (population, intervention, comparison, results, study design):
PopulationChildren and adolescents (1–18 years) with T1D.InterventionInsulin bolus given immediately before meal (START: ‒2 to 0 min) or post-meal (POST: 10–20 min after the start of the meal) Rapid analogue insulin bolus or mealtime (START) or post-meal (POST) fast-acting insulin analogue bolus.ComparisonPre-meal bolus (‒20 to ‒10 min), the gold standard in adults.Outcomes(i) post-prandial glucose levels, blood glucose area under the curve (AUC), maximum blood glucose level; (ii) HbA1c, number of hypoglycemic episodes, diabetic ketoacidosis (DKA) episodes, total daily insulin dose, time in range, time below range; (iii) BMI; (iv) treatment satisfaction.Study designRandomized clinical trials (RCTs), observational studies (cohort, case-control, cross-sectional studies), exploratory studies, mix of qualitative and quantitative studies.

The inclusion criteria in this systematic review included (i) youths aged 1–18 year with T1D; (ii) observational studies, exploratory studies, mix of qualitative and quantitative studies; (iii) we excluded review articles, after their reference lists screening to identify potential eligible studies; (iv) only full text papers were included, whereas abstract only were not included; (v) data on intervention (different timing of pre-meal bolus) (vi) publication date in the last 20 years (1 January 2003–30 September 2022).

Exclusion criteria were: (i) data available only for adults ≥18 years; (ii) case reports; studies with <10 children or adolescents with T1D; (iii) full paper not available; (iv) study not yet published; (v) studies not reporting different timing of bolus dose; (vi) languages other than English were not “a priori” exclusion criteria.

### 2.3. Data Extraction and Management

Two review authors (EM and RF) independently screened for inclusion the title and abstract of all the studies identified using the search strategy, with any disagreement resolved by a third reviewer (MLM). After abstract selection, 4 investigators conducted the full paper analysis.

The following characteristics were reviewed for each included study: (i) reference aspects: authorship(s); published or unpublished; year of publication; year in which the study was conducted; other relevant cited papers; (ii) study characteristics: study design, type of rapid or fast-acting insulin analogue and modality of bolus delivery, timing of bolus; (iii) population characteristics: age, number of pediatric participants with T1D, setting, treatment regimen, meal duration, meal composition, period, region; (iv) methodology: pre-prandial glucose targets, frequency of glucose monitoring, bolus timing: (v) main results: assessment of post-prandial glucose levels, HbA1c, patient and parent’s satisfaction.

### 2.4. Assessment of the Certainty of the Evidence

Grading of recommendations assessment, development and evaluation (GRADE) was used to assess the certainty of evidence (www.gradeworkinggroup.org, accessed on 22 October 2022) for the included studies. GRADE was independently completed by 2 review authors (EM, RF) and the quality of evidence was rated for each of the outcomes above reported. In the case of risk bias in the study design, imprecision of estimates, inconsistency across studies, indirectness of the evidence, and publication bias, the option of decreasing the level of certainty by 1 or 2 levels according to the GRADE guidelines was applied [15]. GRADE has 4 levels of quality of evidence: very low, low, moderate, and high.
HighThe authors have a lot of confidence that the true effect is similar to the estimated effect.ModerateThe authors believe that the true effect is probably close to the estimated effect.LowThe true effect might be markedly different from the estimated effect.Very lowThe true effect is probably markedly different from the estimated effect.

## 3. Evidence from Clinical Studies

The PRISMA flow diagram (Figure 1) summarizes the process of publications screening.

A final number of 13 studies were included in this systematic review. A summary of studies along with the grading of evidence are reported in Table 1 [10,16,17,18,19,20,21,22,23,24,25,26].

According to moderate–high level quality studies, these are the main results:

### 3.1. Glycemic Outcomes

*Post-prandial blood glucose*: Administration of rapid-acting analogues a few minutes before, compared with immediately before or after the meal, leads to a smaller peak blood glucose at one hour after lunch [27] and up to two hours after breakfast [16,17]. No differences in these glycemic parameters were found between children and adolescents [17]. Time to reach post-meal glucose peak was longer when using a pre-meal bolus 20′ before a carbohydrate-rich meal compared with a similar bolus at the beginning of the meal [21]. Only one study reported data on time above range (TAR) and showed that a pre-meal bolus was associated with a better TAR compared with a bolus at the beginning or after the meal [16].

Fast-acting analogues administered at mealtime or post-meal, compared with rapid insulin analogues, provided an additional advantage in terms of reduced post-meal blood glucose peaks in one pediatric study, using fast-acting insulin aspart (FIAsp) [10], and in another using ultra-rapid lispro (URLi) [26]. Blood glucose was lower at one hour post-meal in two studies [24,26], and up to two hours in children, but not in adolescents, with T1D [10].

*HbA1c and hypoglycemia:* Some studies reported a better HbA1c in individuals who injected rapid-acting insulin analogue before a meal compared to immediately before or after, with no differences in risk of hypoglycemia [16,17,22,27], or even a reduced risk as reported in the study by Peters et al. [20]. These data are very important because, particularly in the pediatric population, giving insulin before a meal is associated with parental concerns about risk of hypoglycemia [20]. Another study did not show any difference in HbA1c associated with timing of bolus in children nor in adolescents [17].

In one study, the use of a fast-acting analogue (FIASP) administered at mealtime or post-meal, to cover a standardized liquid meal, compared with pre-meal rapid insulin analogues, led to better HbA1c results without increasing hypoglycemia [24]. However, another study assessing the fast-acting URLi compared with rapid-acting analogue, to cover real life meals, did not show any difference in HbA1c whereas hypoglycemia (blood glucose <54 mg/dL) two hours post-meal was increased [26].

*Other benefits:* A pre-meal standard bolus when eating an Italian ‘‘margherita’’ pizza was associated with a reduced zero to six hour glucose area under the curve, without an increase in hypoglycemia, and no differences in post-prandial blood glucose (PBG) and blood glucose (BG) peak was detected [23]. In an adolescent cohort of adolescents with T1D, pre-meal insulin bolus was associated with a reduced prevalence of missed bolus. This is an important finding given that missing even one meal insulin dose per week is associated with suboptimal glycemia and increased risk of DKA [22].

### 3.2. Total Daily Insulin Dose and BMI

One study with rapid-acting analogues showed that post-meal insulin bolus was associated with a higher total daily insulin dose and BMI compared to a bolus given pre-meal or at the same time as the meal in young people with T1D aged 12–18 years [20]. In contrast, no differences in these parameters were found in another study comparing the fast-rapid-acting analogue FIAsp given at mealtime and post-meal FIAsp with mealtime IAsp [24].

### 3.3. Treatment Satisfaction

Treatment satisfaction was analyzed only in one study, which reported no differences associated with timing of bolus, as well as no differences when analyzing data separately for children and adolescents [17].

## 4. Discussion

This systematic review provides an updated summary of current evidence, graded with the GRADE approach, on timing of insulin boluses in the pediatric population. In 2017, Slattery et al. conducted a systematic review on this topic in adults, and concluded that a rapid-acting insulin analog injected 15–20 min before a meal was associated with a ~30% reduction in post-meal glucose levels compared with when injected immediately before the meal [8]. Moreover, a greater risk of post-prandial hypoglycemia was detected when patients injected rapid-acting analogues after compared with before eating [8].

This systematic review shows that one third of children and adolescents inject rapid-acting analogues after a meal [20,22], despite the ISPAD 2018 recommendation of pre-meal injection [1]. Potential explanations for this observation are that post-meal insulin administration might facilitate a better evaluation of the carbohydrates consumed by the child, reduces parental concerns about the risk of hypoglycemia due to delayed or partial consumption of the meal, and could increase treatment satisfaction [20].

In this review, we analyzed selected studies considering three different bolus timings: pre-meal bolus: −20 to −10 min before the meal; at start of the meal: −2 to 0 min; post-meal bolus: 10–20 min after the start of the meal.

According to moderate–high level quality studies, these are the main findings of this review:Similar to adults, in the pediatric population, individuals using pre-meal insulin injection showed better glycemic outcomes (post-prandial BG, HbA1c, and hypoglycemia) compared with those on post-meal injections.Studies on fast-acting analogues confirmed the feasibility of post-meal dosing, which could contribute to lower BG levels for two hours after the meal according to their pharmacokinetic properties [10].The available data on treatment satisfaction are insufficient to make any conclusion about a negative effect on quality of life associated with pre-meal compared to post-meal bolusing.Only a few studies reported CGM data, which are a very important tool to move towards a personalized approach for the timing of insulin boluses based on individual characteristics, age groups, and meal composition. CGM data also provides valuable information on the individual’s glucose trends (stable, increasing, or decreasing levels) to adapt insulin timing or dose and improve time in range (TIR) [28].

The main strengths of this review are the stringent inclusion criteria, the inclusion of studies covering a 20-year period, the application of the PICOS model for the selection of studies, and the GRADE system for evidence assessment. The limitations are the heterogeneity of the studies, in terms of sample size, age of the study population, and the included outcomes. Due to this heterogeneity, a meta-analysis was not possible. In addition, assessment of the outcomes according to age groups was not possible due to limited data. Another key limitation of the available studies is lack of glycemic outcomes based on CGM systems, which are essential to move towards a personalized timing of boluses according to age groups and individual characteristics. Future studies assessing timing of boluses using CGM are needed.

## 5. Conclusions

The results of this systematic review are in line with those from studies in adults with T1D, in showing that a pre-prandial bolus provided better post-prandial glycemia and HbA1c without increasing the risk of hypoglycemia, and without affecting total daily insulin dose and BMI. For young children who often have variable eating behaviors, fast-acting analogues administered at mealtime or post-meal [29] could provide an additional advantage [24,26].

## Figures and Tables

**Figure 1 jpm-12-02058-f001:**
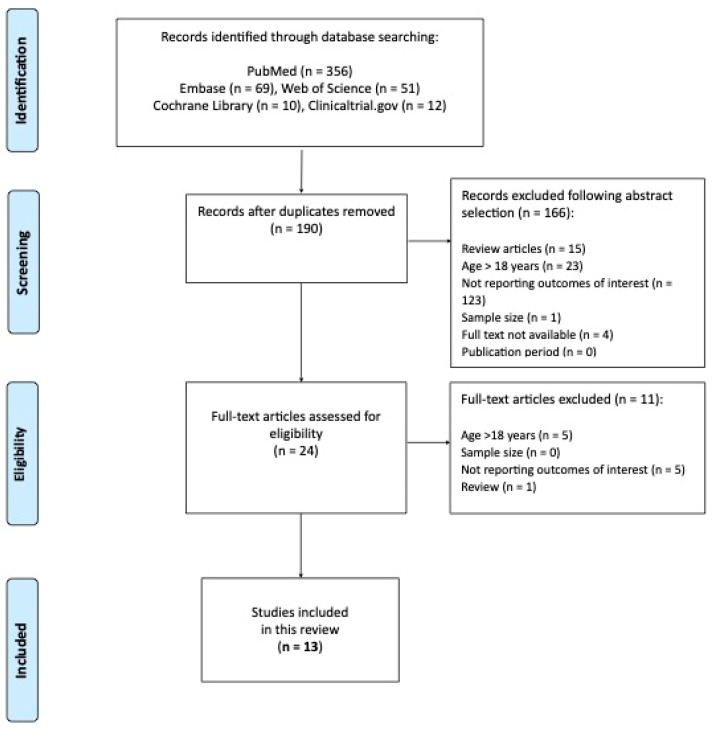
The PRISMA flow diagram.

**Table 1 jpm-12-02058-t001:** Literature analysis after PICOS selection: description, summary of the studies and grading of evidence using the GRADE system. PRE: −20 to −10 min before the meal; START: −2 to 0 min; POST: 10–20 min after the start of the meal.

Rapid-Acting Analogs							
References	Main Objective	Study Design	Population and Comparator, Setting	Methods	Bolus Timing	Results	Study Limitations and Level of Evidence
Scaramuzza AE et al. [27]	Effect of different timing of bolus dose	cross-sectional	30 T1D Age: 6–20 yrs (15.2 ± 3.9) Treatment: CSII, Aspart Setting: hospital for 3 days Period: 2009 Region: Italy	Meal: standard lunch (55% CHO) for 3 days, lasted 15–20 min Pre-prandial BG: 80–140 mg/dL Capillary BG monitor: −15, 0, 30, 60, 90, 120, 180 min Outcome: 1 h- and 3 h-PBG, AUC; number of hypoglycemic events	−15 (PRE), immediately before (START) and immediately after the meal (POST) randomly assigned to each patient	1 h-PBG was lower when bolus PRE or START vs. POST l (*p* = 0.024), not significant at 3 h-PBG No difference in PBG at any time, when bolus was administered PRE vs. START Lower AUC for glycemia with bolus PRE, but NS Hypoglycemia: 12 patients experienced 1 episode each of mild hypoglycemia	-Moderate-
Cobry E et al. (2010) [16]	Determine the optimal timing of insulin bolus delivery	cross-over	23 T1D Age: 12–30 yrs (18.3 ± 4.4; 11 pediatric) Treatment: CSII, Glulisine Setting: 3 clinical visits Period: 2009 Region: Colorado	Meal: frozen prepackaged breakfast Pre-prandial BG: 100–180 mg/dL Capillary BG monitor: 30, 60, 90, 120, 150, 180, 210, 240 min Outcome: 1–2 h PBG, BG peak, TAR, AUC, hypoglycemia	−20 (PRE) immediately before (START) and +20 min (POST), randomized	Lower 1 h- and 2 h-PBG with PRE vs. START (0.0029 and 0.0294) vs. POST (*p* = 0.001 and 0.0408) bolus. No differences between START and POST Lower BG readings above 180 mg/dL in PRE vs. START bolus (*p* < 0.0001) and POST bolus (*p* < 0.0001) Lower AUC with PRE vs. START bolus (*p* = 0.0297) Lower peak BG with PRE vs. START bolus (*p* = 0.0039) and POST bolus(*p* = 0.0027). Hypoglycemia: no significant difference among the different treatment groups	Small pediatric sample -Moderate-
Danne T et al. (2003) [17]	Compare PBG after pre-prandial vs. post-prandial insulin injection	Randomized, open-labeled, cross-over trial 6 weeks period	42 T1D 6–12 yrs 34 T1D 13–17 yrs (12.2 ± 2.8 yrs) Treatment: MDI (long-acting basal insulin: NPH, lente, or ultralente) Aspart Setting: 3 visits in 6 week period Period: 2003 Region: Germany, Austria, Sweden	Meal: all Capillary BG profiles (before, 120 min after meal and at 10:00 p.m. ± 1 h) Treatment Satisfaction Questionnaire (DTSQ) completed by adolescents and parents of the children at the clinic before and after each treatment period. Outcome: 2 h-PBG, Fructosamine (+6 weeks), HbA1c (+6 weeks), hypoglycemia, DTSQ score	Immediately before (PRE), immediately after (0–30′) meal start (POST)	Lower PBG 120 min after breakfast for IAsp PRE vs. POST (*p* = 0.016) Fructosamine and HbA1c: no difference in IAsp PRE vs. IAsp POST The relative risk of hypoglycemia was not significantly different (*p* = 0.31) No clinically relevant differences were found between the two age groups in any of the parameters Treatment satisfaction was equally high for both regimens with both patients and parents	NPH use -Moderate-
Rohilla L et al. (2021) [18]	Real world data on post-prandial bolusing in young children with T1D	Retrospective study	44 T1D Age: 2–7 yrs (4.1 ± 1.3) Treatment: MDI in basal bolus Period: 2015–2021 Region: North India	Meal: all Capillary BG 2 years f/up Outcome: hypoglycemia, DKA, HbA1c	10–20′ before (PRE). during or within 10′ after meal (POST)	HbA1c: no difference during f/up between Group 1 and Group 2 DKA, number of hypoglycemic episodes: not different	PBG not detected. The only study with age <6 y -Low-
Lane W et al. (2021) [19]	Review of the burden associated with pre-meal insulin administration	Prospective online survey	350 parents of children ≤15 yrs Treatment: 70% MDI Aspart and Lispro Period: 2019–2020 Region: USA, Canada, UK, Japan, Spain, and France	Meal: all Online survey Outcome: burden, quality of life	15–20′ before (PRE) 0–2′ before (START) after the start of the meal (POST)	93% of parents felt that PRE bolus has a negative impact on the child’s day to day life Having the freedom to administer insulin at START or POST would have a positive impact	Online survey -Low-
Peters A et al. (2017) [20]	Assess prevalence and characteristics of children and adolescents with T1D using pre-prandial vs. post-prandial bolus	Cross sectional study, data from T1D Exchange registry	21533 T1D (12450 < 18 yrs) Treatment: 99% used rapid-acting insulin. Pump users 48% Period: 2010–2012 Region: USA	Meal: all Capillary BG Survey: when do you usually give an insulin bolus? Outcomes: HbA1c, total daily insulin dose/Kg, hypoglycemia, DKA, BMI	Insulin several minutes before or immediately before meal (PRE) vs. during meal or after meal (POST) = 32%	Children dosing POST (32%) were characterized by higher HbA1c (*p* < 0.0001), larger total daily insulin dose/Kg (*p* < 0.0001), greater prevalence of history of hypoglycemia (*p* = 0.0071) and DKA (*p* = 0.02) vs. PRE BMI was significantly increased in the POST group versus PRE for ages 12–18 yrs only (*p* 0.078)	Cross-sectional design -Moderate-
Tucholski K et al. (2019) [21]	Assess PBG in children and adolescents using CSII after carbohydrate-rich meals	Cross over RCT	29 T1D Age: 9.6–15.2 yrs Treatment: CSII, rapid-acting insulin Period: 2009–2010 Region: Poland	Meal: over a period of 3 days, consumption of carbohydrate-rich meal (60–65%) at breakfast Outcomes: CGM: PBG at 0, 120, 180′, glucose peak, AUC, hypoglycemia	Insulin 20 min before (PRE) vs. 10′ before (PRE) vs. 0′ (START)	Patients who administered bolus 20 min PRE vs. at START had longer median time to reach peak glucose (*p* = 0.01) PBG and peak differences were NS Hypoglycemia: NS	-Moderate-
Datye KA et al. (2018) [22]	Explore the association between timing of insulin bolus and missed bolus	Cross sectional study, data from T1D exchange registry	3608 T1D < 18 yrs Treatment: CSII (60%) Period: 2010–2012 Region: USA	Meal: all Capillary BG Survey on timing of bolus at meal, frequency of missed meal insulin doses Outcomes: prevalence of bolus before meal, population characteristics, missed bolus, HbA1c, hypoglycemia	Several minutes before (PRE), immediately before (START), during-after meal (POST), and “I do not give a mealtime bolus”. Frequency of missed meal insulin doses (from never to once a day)	Prevalence: Insulin PRE (21%), at START (44%), or POST (during 10%, after 24%) Giving insulin PRE or at START was reported by 61% of participants/parents <6 yrs of age, 72% of those 6–13 yrs, 68% of those 13–18 yrs Insulin PRE: usually younger patient, shorter DT1 duration, more likely to use pump therapy, monitored BG more frequently Insulin PRE was associated with lower HbA1c and fewer missed meal insulin doses (*p* < 0.01) (vs. during or after meal). No association between timing of meal insulin and occurrence of severe hypoglycemia events	-Moderate-
**Rapid-Acting Analogs and a Pizza Meal**							
De Palma A et al. (2011) [23]	Evaluate the most effective type and timing of a pump-delivered, pre-prandial bolus for a pizza ‘‘margherita’’ meal	Longitudinal study	38 T1D Age: 6–19 yrs Treatment: CSII, rapid-acting insulin Period: 2010 Setting: hospital Region: Italy	Meal based on pizza Margherita, at lunch Capillary BG −15, 0, +30, 60, 90, 120, 180, 240, 300 Outcomes: BG, hypoglycemia, AUC 0–6 h	(a) a dual-wave bolus 30%/70% over a 6-h period, administered 15 min PRE(b) a dual-wave bolus 30/70% given over a 6-h period, at START; (c) a standard bolus 15 min PRE(d) a standard bolus at START	The simple bolus 15 min PRE, rather than at START or delivered as a double-wave bolus, is better to control the glycemic rise (AUC 0–6 h) usually observed (*p* < 0.01) No difference in hypoglycemia	-Moderate-
**Fast-Acting Analogs**							
Bode B et al. (2019) [24]	Assess the efficacy and safety of faster aspart vs. IAsp	RCT	777 patients with DT1 < 18 yrs Treatment: FAsp vs. IAsp for 26 weekswith Degludec Period: 2016–2018 Region: 150 sites across 17 countries	Meal: a standardized liquid meal at main meals Capillary BG. CGM in a subgroup of 135 patients Outcomes: HbA1c, hypoglycemia, PBG at 1 h, TDI	260 mealtime FAsp 258 mealtime IAsp 269 post-meal FAsp	HbA1c: Mealtime and post-meal FAsp performed better than IAsp (*p* = 0.014) Lower 1-h PBG increment with FAsp versus IAsp over all meals (*p* < 0.01 for all) No significant differences in the overall rate of hypoglycemia, severe hypoglycemia, insulin dose and BMI	Home-sampling kit to measure FPG -Moderate-
Kawamura T et al. 2021 [25]	Assess the efficacy and safety of faster aspart vs. IAsp	Post-hoc subgroup analysis based on data from the RCT onset 7 trial	66 T1D < 18 yrs Treatment: FAsp vs. IAsp for 26 weeks with Degludec Period: 2013–2015 Region: Japan	Meal: all Capillary BG profiles at baseline and week 26 Follow-up on day 7 and day 30 Pre-prandial target BG: 71–145 mg/dL;Bedtime 120–180 mg/dL Outcomes: HbA1c, PBG, hypogycemia, insulin dose, body weight	24 mealtime FAsp 19 post-meal FAsp 23 mealtime IAsp	HbA1c: the post-prandial FAsp performed better (with a change from baseline of 0.74%) than the meal FAsp (0.23%) and IAsp (0.39%) Lower 1-h PBG increment with mealtime FIAsp versus IAsp over all meals No differences in the overall rate of hypoglycemia, severe hypoglycemia, insulin dose	Low sample size, which precluded statistical analysis between the treatment groups -Low -
Fath M et al. 2017 [10]	Assess FIASP exposure and action in children and adolescents vs. IAsp	RCT	12 children with T1D (6–11 yrs) 13 adolescents with T1D (12–17 yrs) Treatment: MDI and CSII; FiAsp vs. IAsp Period: 2014 Region: Hanover (Germany)	Meal: a standardized liquid meal (BOOST, Nestlé S.A) consumed within 8 min. The volume of the liquid meal was adjusted according to the subject’s body weight Two dosing visits and a follow up visit. At each dosing visit, a stable glucose level was achieved overnight using an established protocol of variable intravenous infusion Capillary BG Outcomes: PBG from 0 to 2 h	Subjects received 0.2 U/kg subcutaneous dosing immediately prior to a standardized meal	Onset of appearance occurred 5–7 min earlier and exposure was greater for FIASP vs. IAsp in children and adolescents PG excursion appeared to be reduced for faster aspart compared with IAsp at 0–1 h (*p* = 0.05) and at 0–2 h (*p* = 0.028) and as peak (*p* = 0.044) in children but not in adolescents	Low sample size, standardized liquid meal -Moderate-
Wadwa RP et al. 2022 [26]	Assess the efficacy of ultra-rapid lispro (URLi) versus lispro	RCT prospective, double-blind	716 T1D Age 12.26 ± 3.39 yrs Treatment: MDI Period: 2019–2021 Region: USA, 96 sites	Meal: all 26-week treatment period: randomized to double-blind and pre-study basal insulin Capillary BG and CGM systems Outcomes: HBA1c, PBG, insulin dose, hypoglycemia	URLi (n = 280) or lispro (n = 298) Injection 0–2 min prior to meals (mealtime) vs. open-label URLi (n = 138) injected up to 20 min after start of meals (post-meal)	HbA1c: no significant differences among the treatment groups after 26 weeks When dosed at the beginning of meals, URLi reduced 1-h PBG (*p* = 0.001) and PPG excursions versus lispro (*p* < 0.001) Hypoglycemia: mealtime URLli vs. Lispro presented higher rate of hypoglycemia (<54 mg/dL) at ≤2 h (*p* = 0.034) CGM group (n = 79): no difference in AUC 0–2 h	Poor CGM data –high

T1D: type 1 diabetes, min: minutes, yrs: years, BG: blood glucose, PBG: post-prandial BG, MDI: multiple daily injection, CSII: continuous subcutaneous insulin infusion, HbA1c: hemoglobin A1c, CGM: continuous glucose monitoring, FGM: flash glucose monitoring, FIAsp: faster insulin aspart, URLi: ultra-rapid lispro.

## Data Availability

All databases generated for this study are included in the article.

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
