# Peer review of "Optimal Prandial Timing of Insulin Bolus in Youths with Type 1 Diabetes: A Systematic Review"

_jpm, 2022, doi:10.3390/jpm12122058_

Round 1

Reviewer 1 Report

The following Changes are desired:

1. All headings should be in title case.

2. Figure quality need improvement so that the text is easily readable.

3. The mathematical terms/symbols used in full manuscript should me math case (in $symbol$) including the figure.

Author Response

I am grateful for the careful review and the helpful comments and suggestions. We considered all comments and addressed them point by point below. All changes in the revised manuscript are highlighted accordingly in a track-changed version.

  1. All headings should be in title case.

    Following the reviewer’s suggestion, we have changed all headings to title case.
  1. Figure quality need improvement so that the text is easily readable.

Thanks, we added a new Figure in .jpeg format, converted from the original .doc one

  1. The mathematical terms/symbols used in full manuscript should me math case (in $symbol$) including the figure.

I apologize, but I can't understand this question. Do you mean to change  "< 18 years " to "less than 18 years" and so on?

Reviewer 2 Report

1. Include conclusion and implications of the findings at the end of the abstract. 

2. There is no discussion in the main text. Discuss the findings in light of the existing literature and their implications in the main text. 

3. Why a meta-analysis was not performed?

Author Response

I am grateful for the careful review and the helpful comments and suggestions. We considered all comments and addressed them point by point below. All changes in the revised manuscript are highlighted accordingly in a track-changed version.

  1. Include conclusion and implications of the findings at the end of the abstract. 

As suggested we have added some conclusion and implications at the end of the abstract

  1. There is no discussion in the main text. Discuss the findings in light of the existing literature and their implications in the main text. 

We have now added a discussion section. In the previous version of the manuscript, we had merged results and discussion.

  1. Why a meta-analysis was not performed?

Thanks, we added in the discussion section that because of heterogeneity of study populations, study design and measured outcomes, results could not be combined in a meta-analysis.